# Effect of Nanocellulose on the Properties of Cottonseed Protein Isolate as a Paper Strength Agent

**DOI:** 10.3390/ma14154128

**Published:** 2021-07-24

**Authors:** Jacobs H. Jordan, Huai N. Cheng, Michael W. Easson, Wei Yao, Brian D. Condon, Bruce C. Gibb

**Affiliations:** 1The Southern Regional Research Center, Agricultural Research Service, USDA, 1100 Robert E. Lee Blvd., New Orleans, LA 70124, USA; jacobs.jordan@usda.gov (J.H.J.); hn.cheng@usda.gov (H.N.C.); brian.condon@usda.gov (B.D.C.); 2Department of Chemistry, 2015 Percival Stern Hall, 6400 Freret Street, Tulane University, New Orleans, LA 70118, USA; Wyao1@tulane.edu (W.Y.); bgibb@tulane.edu (B.C.G.)

**Keywords:** cottonseed protein, cellulose nanocrystals, cellulose nanofibers, paper, dry strength

## Abstract

Currently, there is an increasing interest in the use of biopolymers in industrial applications to replace petroleum-based additives, since they are abundantly available, renewable and sustainable. Cottonseed protein is a biopolymer that, when used as a modifier, has shown improved performance for wood adhesives and paper products. Thus, it would be useful to explore the feasibility of using cellulose nanomaterials to further improve the performance of cottonseed protein as a paper strength agent. This research characterized the performance of cottonseed protein isolate with/without cellulose nanofibers (CNFs) and cellulose nanocrystals (CNCs) to increase the dry strength of filter paper. An application of 10% protein solution with CNCs (10:1) or CNFs (50:1) improved the elongation at break, tensile strength and modulus of treated paper products compared to the improved performance of cottonseed protein alone. Further analysis using scanning electron microscopy (SEM) and Fourier transform infrared spectroscopy (FTIR) indicated that the cottonseed protein/nanocellulose composites interacted with the filter paper fibers, imparting an increased dry strength.

## 1. Introduction

Paper is a ubiquitous material used for a large variety of applications including printing, writing, tissues, towels, newsprint, packaging and paperboard. Because of its varying applications, additives are often needed in order to enhance its properties [1,2,3]. Common types of paper additives comprise specific agents that improve the paper’s dry and wet strength. These include cationic starch and acrylamide for dry strength [4], and formaldehyde resins, poly(amino-amide)-epichlorohydrin resins and polyacrylamide polymers for wet strength [5]. In view of the increasing need for sustainability and awareness of microplastic contaminations, more biodegradable and less toxic paper additives are desirable [3].

Nanocellulose is obtained from mechanical or chemical treatments of cellulose fibers to produce either cellulose nanofibers (CNFs) or cellulose nanocrystals (CNCs), collectively cellulose nanomaterials (CNMs) [6,7]. Both CNCs and CNFs have found use as a reinforcement material in polymers, films, gels, foams and composites [8,9,10,11,12]. Unlike petroleum-based fibers, CNCs and CNFs do not persist for extended periods of time in the environment and are completely biodegradable [13,14]. Carboxymethylated CNFs improved the wet and dry performance of laboratory-prepared paper sheets when used as an additive with poly (aminoamide)-epichlorohydrin [15]. Periodate oxidation of CNCs containing sulfate half-esters significantly improved the dry and wet strength of paper products at a dosage of 0.05–1.2 wt % [16]. Unmodified CNCs containing only sulfate half-esters and no carbonyl groups performed similarly in dry strength tests but did not improve paper’s wet tensile strength. The authors attributed this difference to the ability of carbonyl-containing CNCs to form covalent bonds between cellulose chains. Additionally, nanocellulose was used as a reinforcement material for recycled paper [17].

Cottonseed protein (CSP) have been used for the formulation of biobased products [18,19,20], as animal feed [21,22], in films, coatings and adhesives [19]. Several proteins have been previously reported as binders and strength agents in paper products, such as soy protein [23,24,25,26,27], gelatin, zein, and hydroxyproline-rich glycoprotein [28] and cottonseed protein [29]. Related work has shown that CSP isolate served as an effective strength agent for nonwovens [30]. In an adhesive study, improved performance was observed when CNCs and CNFs were used as strength additives with CSP [31]. Although both CSP and nanocellulose are “green” and available, thus far, no one has combined both CSP and nanocellulose as strength agents for paper. The goal of this work is to show that the performance of CSP as a strength agent for paper products will be improved by the use of nanocellulose.

## 2. Materials and Methods

### 2.1. Materials

Reagents were purchased from commercial suppliers Millipore Sigma (St. Louis, MO, USA) or VWR (Radnor, PA, USA), unless otherwise stated. Cotton gin motes (CGM) were supplied by the USDA research facility in Stoneville, MS, USA. Cottonseed protein was prepared from the defatted seed of the glandless cottonseed by the base solubilization and the acid precipitation procedure previously reported [32,33].

### 2.2. Preparation of Nanocellulose

To prepare nanocellulose suspensions, cellulose was first isolated from CGM by the procedure previously reported [34]. Briefly, CGM were mechanically ground to <20 mesh with a Wiley mill (E3300, Eberbach Corp., Belleville, MI, USA). The obtained dry powder was treated for 2 h at 70 °C with a 4% (*w/w*) sodium hydroxide (NaOH) solution at a fiber to liquor ratio of 1:20 (*w/v*). Afterwards, the fibers were washed with deionized water until a pH ≈ 6–7 eluant was achieved. The remaining fibers were then exhaustively bleached (fiber to liquor ratio, 1:20 (*w/v*)) three times at 75 °C for 2 h with an acidified sodium chlorite (NaClO_2_, 0.50%, *w/v*) solution containing 1.0% acetic acid (*v/v*). The cellulose was recovered after thoroughly washing it with deionized water and then dried to a constant mass at 70 °C.

#### 2.2.1. Preparation of Cellulose Nanocrystals

The CNCs were prepared as previously described, using sulfuric acid (H_2_SO_4_) hydrolysis of isolated cellulose for 60 min at 60 °C and 62% (*w/w*) H_2_SO_4_ at a material to liquid ratio of 1:20 (*w/v*) [35]. The suspension was quenched with ice water and then washed by successive centrifugation cycles at 16,000× *g* for 15 min each cycle until a turbid supernatant was obtained. The CNCs we obtained were dispersed using a 750 W ultrasonic processor (Vibra-Cell probe sonicator, VCX-750, Sonics & Materials, Newton, CT, USA) with a 60% power amplitude for 5 min. Large particulates were removed by centrifugation (3500× *g*, 5 min) and the CNCs were purified by dialysis against deionized water until the conductivity stabilized below 2 μS·cm^−1^ for two successive bath changes. For the dialysis, a regenerated cellulose dialysis tube (MW cut-off 10,000) was used. The resulting CNC suspension was stored in refrigeration (4–8 °C).

#### 2.2.2. Preparation of Cellulose Nanofibers

The CNF suspensions were prepared by the procedures previously described, using wet-disk milling and high-pressure homogenization [36]. Briefly, 50 g of isolated cellulose was suspended in 3 L of deionized water using an Ultra-Turrax^®^ (T25, IKA Works, Inc., Wilmington, NC, USA) mechanical homogenizer to obtain a 2% (*w/w*) cellulose slurry. The slurry was passed through a Supermasscolloider MKCA6-2 (Masuko Sangyo Co., Ltd., Saitama, Japan) for ten successive passes, defibrillating the pulp by high-shear forces [9]. For each pass, the disk clearance was maintained at 4 µm and the rotational speed was 12,000 rpm. The nanocellulose slurry was then subjected to high-shearing forces using a high-pressure homogenizer (Microfluidizer M-110EH, Microfluidics Corp., Newton, MA, USA). The slurry was injected, and the suspension was pumped through one 200 µm ceramic and one 87 µm diamond Z-shaped interaction chamber for a total of five passes. After the suspension was collected, the concentration was adjusted as necessary, prior to analysis.

### 2.3. Paper Analysis

The paper used was Whatman #1, manufactured from cotton linters with a manufacturer-specified thickness of 180 µm. The analysis of dry paper strength was adapted from ASTM D 828-97. Paper sheets were cut into 2.54 cm × 15.24 cm strips. For the removal of water-soluble contaminants, the strips were immersed in deionized water and allowed to air dry under ambient conditions. Strips were coated with a solution of cottonseed protein (10% w/w, pH 10.5) or cottonseed protein in a nanocellulose slurry. For all of the formulations, ultra-pure water with a minimum resistance of 18.2 MΩ was used. At 25 °C, conductivity was 0.055 µS/cm.

Each dispersion (CSP + CNM) was stirred for at least one hour and then homogenized with a mixer. The pH values of all the dispersions were adjusted to pH of 10.5 by the addition of small aliquots of concentrated NaOH. The dry weights of the paper before and after the addition of modifiers (CSP and CNM) were measured.

Each formulation was applied to seven paper strips using a soft brush. Control formulations—adjusted to pH 10.5—containing only dilute NaOH (~0.5 mM), CNCs (1.0 wt %), or CNFs (0.2 wt %) were similarly applied. The treated strips of paper were then dried under ambient conditions. The dried strips were heat-pressed (120 °C, 0.25 MPa, 10 min) using a heated benchtop press (Model 3856, Carver Inc., Wabash, IN, USA). Paper strips were weighed prior to and following the application of the protein formulations, and the paper thickness of the final sample was measured with a digital precision thickness gauge (FT3, Hanatek Instruments, UK). Characterization of prepared paper strips and formulations was performed via thermogravimetric analysis (TGA), Fourier transform infrared spectroscopy (FTIR), and scanning electron microscopy (SEM).

The dry and wet paper strengths were measured with a Zwick stress tester (Zwick GmbH & Co., Ulm, Germany). During the analysis, the crosshead speed was 1 mm·min^−1^. During dry strength testing, data was collected for the tensile modulus, the tensile strength, and the maximum elongation at break. For the wet strength analysis, the tensile modulus, the yield strength, and the elongation at the yield point were determined. Analysis of wet paper strength was performed using a method adapted from ASTM D 829-97 [29,37]. The treated paper strips were prepared as described, and then immersed in distilled water for 1 h at 23 °C; immediately after saturation in the water, the strips were tested for strength. For each formulation, seven paper strips were analyzed and the tests for each formulation repeated in (at least) triplicate. Differences in the paper’s mechanical properties were determined using an analysis of variance (ANOVA) and a Tukey means comparison test (α = 0.05).

### 2.4. Fourier Transform Infrared Spectroscopy (FTIR)

The FTIR spectra of paper samples and CSP were obtained using a Bruker Alpha Platinum single reflection Attenuated Total Reflectance (ATR)FTIR spectrometer, equipped with a diamond crystal. Experiments were conducted in transmittance mode under ambient conditions. The samples were pressed into the sample compartment of the ATR spectrometer and 64 scans were acquired with 4 cm^−1^ resolution for each sample in the range from 4000 to 400 cm^−1^.

### 2.5. Scanning Electron Microscopy (SEM)

The surface morphology of control- and treated-paper samples were examined using a field emission scanning electron microscope (FE SEM, Hitachi 4800, Tokyo, Japan). Samples were sputter-coated with a thin layer of carbon for 15 min using a vacuum sputter coater. The data were collected at an acceleration voltage of 3 keV and a beam current of 0.5 nA.

### 2.6. Thermogravimetric Analysis (TGA)

Thermogravimetric (TG) and differential thermogravimetric (DTG) analyses were performed under a nitrogen atmosphere using a TGA Q500 thermal gravimetric analyzer (TA Instruments, New Castle, DE, USA). The nitrogen flow into the furnace was maintained at a rate of 90 mL·min^−1^. A small square, approximately 4–6 mg, was cut from the paper samples and placed into a platinum crucible. The samples were heated from 30 ± 5 °C to 600 °C, with a heating rate of 10 °C min^−1^. The obtained thermogram traces (TG and DTG) were analyzed with Universal Analysis 2000 software (TA Instruments—Waters, LLC, New Castle, DE, USA v4.5A). Each sample analysis was performed in triplicate. The curves were averaged, and the resulting curves plotted with OriginPro 2018b software (OriginLab Corp., Northampton, MA, USA, v9.5).

### 2.7. Differential Scanning Calorimetry (DSC)

Samples of CSP were prepared at 2 mg/mL stock solution and diluted to the desired concentration using ultra-pure water with a minimum resistance of 18.2 MΩ. The pH of the solution was then adjusted to a pH of 10.5 by small aliquots of the NaOH solution. All protein solutions were loaded into the sample cell (~250 μL) with ultra-pure water in the reference cell. All experiments were collected between 40–110 °C at a 1.5 °C·min^−1^ temperature ramp under 50–60 psi. Each experimental result was the average of two measurements and processed with the instrument-packaged software and analyzed using OriginPro 2018b.

## 3. Results & Discussion

### 3.1. Characterization of Paper Samples

The control paper group and the paper samples, treated with CNM dispersions without CSP, essentially showed no change in thickness compared to the untreated filter paper (Table 1); differences in paper thickness between the groups were negligible and considered insignificant. Notably, the control paper was measured at 184.56 ± 14.02 µm, which agrees with the manufacturer specifications of a 180 µm thickness. The application of CSP and CSP with CNM gave average values of approximately 206–212 µm, with a coefficient of variation of 9—11%. This is similar to the variation observed in the thickness of the control paper (9%). Therefore, the application of the CSP and CSP-CNM formulations produced variations in thickness that were reproduceable and did not significantly vary from the original paper product.

For the dry weight add-on, all of the samples treated with CSP showed a significant weight pick-up after drying compared to the control paper. There was negligible change in the dry weight of the paper products after the application of the control formulation, compared to the CNC or CNF formulations. The difference in weight pick-up between CSP and CSP-CNM formulations was considered insignificant. The weight percentage of protein applied onto the paper varied from 18–26%, with the average of each sample approximately being 22%. An application of CSP with CNC or CNF slurry gave the same final paper thickness and weight pick-up with values of approximately 26%, though it was not considered significantly different. Hence, any significant changes in other measured physical properties such as the modulus, or tensile strength, can be attributed to the nanocellulose formulations, and not to the variations in the amount of each formulation applied or the dimensions of the paper samples.

Cottonseed protein isolate is a stable mixture of proteins, measured by melting temperature (Tm), with an observed Tm of >80 °C, pH 10; CSP unfolds reversibly and above this temperature can be entangled and cured to a rigid polymeric substrate, which has seen use in adhesive applications [31,32,33,38]. The DSC data for CSP in solution was obtained (Figure 1). The data showed a Tm of 82 °C, which is expected for CSP isolate. The DSC trace was essentially unperturbed upon the addition of 10 wt % CNC or 2.0 wt % CNFs (relative to CSP) to the suspension, indicating that the (low) concentration of the CNM did not have an effect on the melt transition of CSP, and correspondingly, any reinforcement improvements came from structural and physical reinforcement of the protein coating, and not from a lowering of the Tm.

The thermogravimetric curves and first-derivative thermograms are shown in Figure 2. The TGA and DTG traces for paper and paper treated with CNMs but not CSP are very similar, as are all the samples treated with CSP with and without the presence of CNM. The trace of CSP has a low T_onset_ and produces significant (>30%) char residue. CSP was shown to thermally degrade over a broad temperature range, with an initial T_onset_ of ~280 °C, consistent with prior results [29]. However, the filter paper and paper treated with only cellulose nanomaterials exhibit a sharp thermal degradation with a T_onset_ of ~340 °C, which is expected given their entirely cellulosic composition (See Table 2, for details) [39,40].

The application of the CNM alone had no effect on T_onset_, or T_max_, and was identical between the paper and the paper treated with CNFs. Interestingly, the paper treated with CNCs or treated with CNFs were not significantly different, although, in terms of absolute char residue, the paper treated with CNCs was different than the paper alone. This can be ascribed to the greater concentration of CNCs used in the treatments compared to CNFs, and the breakdown of the sulfate half-esters on the CNC surface and deposition of sodium sulfate salts onto the paper substrate, which provided a modest but significant increase in about 1% greater char residue. A similar trend was observed for paper treated with CSP and CSP with CNCs or CNFs, where the application of CNFs and then CNCs resulted in even greater char-residue for the same reasons outlined above. The change in T_onset_ and T_max_, upon the addition of CNCs compared to CSP without CNCs, again can be attributed to the surface sulfate groups of sulfate CNCs, which have previously been shown to be less stable under thermal heating [35].

To confirm the interaction of CSP and CNM with the paper substrates, FTIR analysis (Figure 3) was performed on the filter paper, paper treated after rinsing with water, and then after treatment with pH 10.5 water, CNCs or CNFs (Figure 3a). In each instance, the FTIR spectra showed characteristic patterns associated with the cotton linters present in the paper samples [41,42]. Specifically, the peaks located at 3100–3600 cm^−1^ and ca. 2900 cm^−1^ correspond to O-H and C-H stretching, respectively, in cellulose (paper and nanocellulose). Additional peaks are indicated in Figure 3c. The peak at 1635 cm^−1^ comes from water that is absorbed on cellulose. The peaks at 1428 cm^−1^, 1367 cm^−1^, and 1334 cm^−1^ correspond to CH_2_ and CH stretching and bending vibrations. The peaks at 890–1060 cm^−1^ are caused by the C–O and C–O–C vibrations in cellulose. The peak at 1420–1430 cm^−1^ is attributed to crystalline cellulose, and the peak at 898 cm^−1^ to the amorphous region of cellulose. Figure 3b shows the FTIR spectra for CSP isolate, as well as the paper samples after the addition of CSP. Since all these peaks do not change with the addition of CSP and nanocellulose, the cellulose has not significantly changed as a result of these additions. Upon the addition of CSP, two prominent absorption bands occur at 1534 cm^−1^ and 1641 cm^−1^, which correspond to the broad absorption from the amide I and amide II bands found in CSP, indicating the CSP was deposited onto the paper substrate [29]. Close examination of the amide I band (Figure 3d) indicates a shift from 1633 cm^−1^ for CSP by itself to 1641 cm^−1^ for the protein in contact with cellulose paper. This 8 cm^−1^ shift likely suggests hydrogen-bonding interactions between the carbonyls of the amide bonds in cottonseed protein and the hydroxyl groups on the cotton fibers [43,44,45]. Hydrogen-bonding between the amide I carbonyls and cellulose substrates has been previously observed between cellulose and cottonseed protein [30,31] and between nanocellulose and amide carbonyls of polyacrylamide [46]. This hydrogen-bonding could potentially account for some of the increased strength observed in the treated samples (vide infra) [30].

Examination of the SEM micrographs (Figure 4) indicates a loose network of randomly distributed fibers for the filter paper structure with numerous void spaces. The application of CNM did not alter this structure; on the higher magnification micrographs of filter paper, treated with CNCs, small rod-like protuberances can be seen on the surfaces of the cotton fibers. In the samples treated with CSP, the small voids are filled, and the protein can be seen to effectively coat the cotton fibers. The application of CSP with CNM (both CNC and CNF) leads to a more uniform coating and, especially for the case of CNFs, filling in much of the interfibrillar spacing, leading to a smoother appearance.

### 3.2. Paper Analysis

#### 3.2.1. Paper Dry Strength Analysis

The results of dry-strength tensile testing of paper samples are reported in Table 3 for tensile modulus and tensile strength testing. Paper treated with the control formulations of only alkali water, or with either of the cellulose nanomaterials, showed no change in mechanical properties where the tensile modulus (0.74–0.77 GPa), tensile strength (9.39–9.70 MPa), and maximum elongation (2.41–2.67%) were all identical. For both the tensile modulus and the tensile strength, the application of CSP improved the mechanical performance in tensile testing by 87% from 0.76 GPa to 1.42 GPa. The tensile strength was similarly improved by 97% from 9.7 MPa to 19.10 MPa. These values are the same order of magnitude as those previously reported for Whatman #1 filter paper, although the improvement is roughly 2-fold greater. This can be attributed to the solution dispersion of cottonseed protein applied at pH 10.5, where the protein was fully soluble, allowing the protein to more effectively penetrate the paper fibers; whereas at pH 4.8, the protein is a dispersion and was shown to be strongly surface adsorbed.

The concomitant application of CNCs resulted in approximately a 20% improvement in the modulus to 1.69 GPa and tensile strength improved 15% to 22.1 MPa. By comparison, the application of the CSP-CNF formulation saw a similar improvement of 20% in the modulus (to 1.65 GPa) and a greater change in the tensile strength (to 23.3 MPa). However, the results between formulations using CNCs, or CNFs, were not significantly different. It is important to note, however, that the CNFs were applied at a much lower (0.2% w/w) concentration (1:50 relative to cottonseed protein), whereas the CNCs were applied at a ratio of 1:10 compared to cottonseed protein.

Paper treated with CSP has a reduced maximum elongation at break (2.30%) compared to plain paper (2.67%). In contrast, the application of only CNCs or CNFs does not significantly change the elongation at break of the plain paper control (2.41%, and 2.57%, respectively). Furthermore, the application of the CSP formulation using CNCs further reduced the elongation at break to 2.13%, which was not significantly different to the application of only CSP. However, the CSP formulation with CNCs was significantly lower than all other treatments and controls. This is in contrast to the CSP formulation using CNFs, where the elongation at break improved to 2.57%, which was identical to the control groups. This indicates the addition of CNFs to the CSP formulation can mitigate the increased stiffness and reduced elongation at break of paper treated with CSP. In addition to hydrogen bonding between amide I carbonyls and hydroxyl groups of the paper, electrostatic interactions between CSP and paper may play a role in improved strength. CSP contains 11–12% arginine, which is cationic at most pH values. In contrast, the paper surface tends to be anionic, thus, it is expected that there will be additional Coulombic interactions between CSP and paper.

#### 3.2.2. Paper Wet Strength Analysis

The data showed that for wet testing, there was no significant difference between the samples treated with CSP and any combination of cellulose nanomaterials (Table 4). This agrees with a previously reported use of CSP as a strength agent for paper products, which did not show improvement in wet-strength testing when combined with paper strength additives, namely, aspartic, adipic, and citric acid [29]. It should be noted that in this report, the application of CSP (at pH 10.5) reduced the tensile modulus and yield strength of the paper samples during wet testing, but significantly improved the yield strain from ~2.5% to ~4.0%.

## 4. Conclusions

This research has shown that cottonseed protein continues to have a broad range of applications in films, coatings, adhesives and paper products, both by itself and when combined with additional fillers such as cellulose nanocrystals and cellulose nanofibers. The application of cottonseed protein isolate with nanocellulose dispersions was shown to improve paper dry-strength performance, improving the tensile modulus and tensile strength of treated paper by 86% and 97%, respectively. No significant changes were observed in paper wet-strength performance compared to the application of cottonseed protein alone. This work suggests that the use of nanocellulose is a promising supplement to cottonseed protein for reinforcement of paper products. Finally, the combination of CSP and CNM offers the consumer an environmentally friendly, biodegradable product choice that is composed entirely of renewable natural resources that do not persist for an extended period of time in the environment.

## Figures and Tables

**Figure 1 materials-14-04128-f001:**
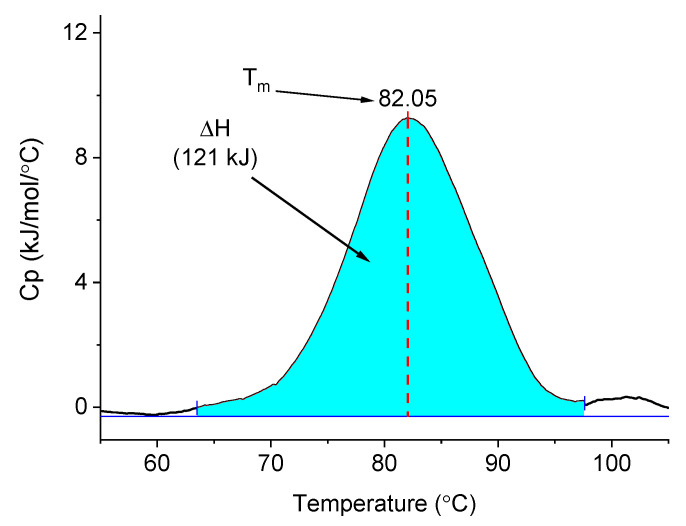
DSC data for CSP obtained at pH 10.5 in aqueous solution.

**Figure 2 materials-14-04128-f002:**
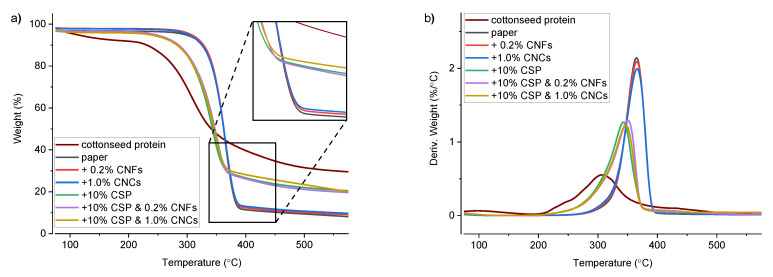
(**a**) TGA and (**b**) DTG data for CSP (brown), paper (black), paper + CNFs (red), paper + CNCs (blue), paper + CSP (green), paper + CSP + CNF (violet), and paper + CSP + CNC (gold); DTG, derivative thermogravimetry; TGA, thermogravimetric analysis.

**Figure 3 materials-14-04128-f003:**
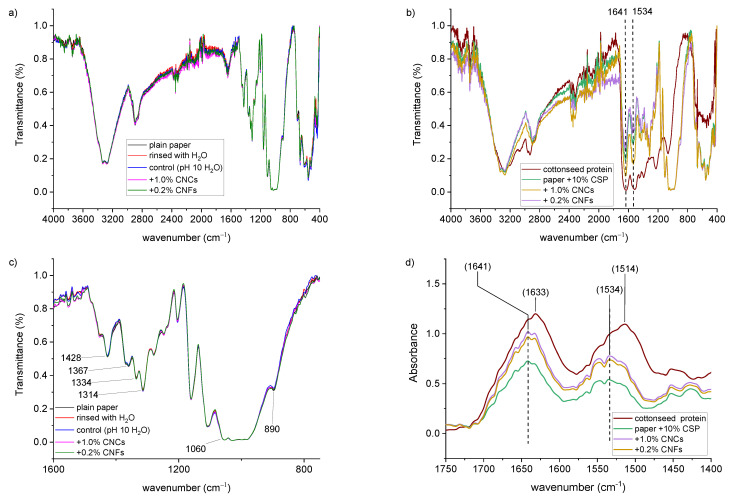
FTIR transmittances spectra of (**a**) paper and control paper sample rinsed with water, treated with pH 10 water, and nanocellulose dispersions; (**b**) FTIR transmittance spectra of cottonseed protein isolate and paper samples treated with cottonseed protein or cottonseed protein and nanocellulose; (**c**) expansion of the cellulosic region from 1600 cm^−1^ to 750 cm^−1^ from (**a**); (**d**) FTIR absorbance spectra of the region about the amide bonds (1750 cm^−1^ to 1400 cm^−1^) from (**b**).

**Figure 4 materials-14-04128-f004:**
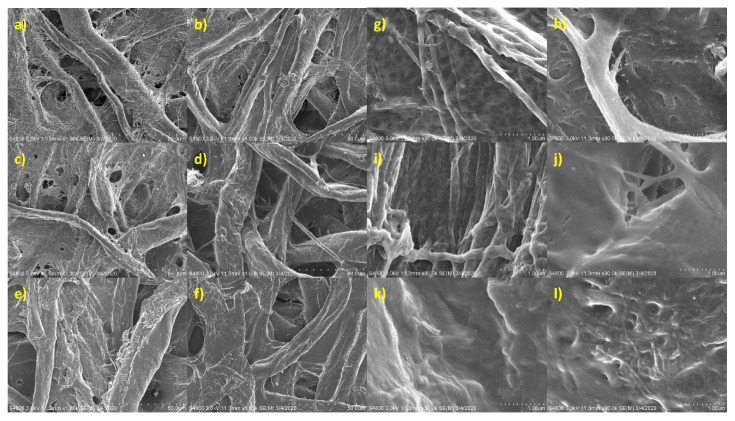
FESEM micrographs at 1000× (**a**–**f**) and 30,000× (**g**–**l**;) of paper (**a**,**g**) and paper treated with: 1 wt % CNC dispersion (**b**,**h**); 0.2 wt % CNF suspension (**c**,**i**); 10 wt % CSP solution (**d**,**j**); 1 wt % CNC dispersion with 10 wt % CSP (**e**,**k**), and; 0.2 wt % CNF dispersion with 10 wt % CSP (**f**,**l**).

**Table 1 materials-14-04128-t001:** Physical properties of paper samples treated with cottonseed protein and nanocellulose formulations.

Sample	Thickness (µm)	Weight Pick-Up (%)
Paper	184.56 ± 14.02 ^a^	1.2 ± 2.0 ^a^
Paper & 0.2% CNF	193.24 ± 13.59 ^a^	2.8 ± 1.7 ^a^
Paper & 1.0% CNC	192.89 ± 18.69 ^a^	3.4 ± 2.5 ^a^
Paper & 10% CSP	209.97 ± 23.89 ^b^	22.4 ± 6.2 ^b^
+0.2% CNF	206.34 ± 18.51 ^b^	26.1 ± 4.6 ^b^
+1.0% CNC	212.49 ± 20.92 ^b^	25.6 ± 7.5 ^b^

* Data with the same superscript letter (a or b) indicate the treatments are not significantly different at *p* < 0.05.

**Table 2 materials-14-04128-t002:** Thermal properties of cottonseed protein isolate and paper samples treated with cottonseed protein and nanocellulose formulations.

Sample	T_onset_	T_max_	Char
CSP	262.5 ± 0.5	307.2 ± 0.3	31.50 ± 0.41
Paper	340.7 ± 0.7 ^b^	364.9 ± 1.2 ^b^	9.39 ± 0.31 ^b^
Paper & 0.2% CNF	339.8 ± 0.1 ^b^	365.5 ± 1.0 ^b^	10.07 ± 0.54 ^b,c^
Paper & 1.0% CNC	338.7 ± 0.1 ^b^	366.0 ± 0.0 ^b^	10.73 ± 0.02 ^c^
Paper & 10% CSP	314.0 ± 2.1 ^c^	351.6 ± 1.6 ^c^	20.18 ± 0.69
+0.2% CNF	311.7 ± 2.0 ^c,d^	350.3 ± 1.4 ^c,d^	22.20 ± 0.13
+1.0% CNC	309.3 ± 2.5 ^d^	347.6 ± 2.5 ^d^	23.40 ± 0.80

* Data with the same superscript letter (a, b, c, or d) within a column indicate the treatments are not significantly different at *p* < 0.05.

**Table 3 materials-14-04128-t003:** Mechanical properties of paper impregnated with cottonseed protein and nanocellulose dispersions.

Sample	Tensile Modulus	Tensile Strength	Elongation @ Break
	(GPa)	(MPa)	(%)
Paper	0.76 ± 0.15 ^a^	9.70 ± 2.13 ^a^	2.67 ± 0.20 ^a^
Paper & 0.2% CNF	0.74 ± 0.14 ^a^	9.70 ± 1.56 ^a^	2.57 ± 0.52 ^a,b^
Paper & 1.0% CNC	0.77 ± 0.13 ^a^	9.39 ± 1.21 ^a^	2.41 ± 0.22 ^a,c^
Paper & 10% CSP	1.42 ± 0.27 ^b^	19.10 ± 1.47 ^bb^	2.30 ± 0.55 ^b,c,d^
+1.0% CNC	1.69 ± 0.21 ^c^	22.06 ± 2.16 ^c^	2.13 ± 0.18 ^c^
+0.2% CNF	1.65 ± 0.28 ^c^	23.34 ± 3.53 ^c^	2.57 ± 0.37 ^a,d^

* Note: Each data point represents seven strips tested in triplicate (21 total tests per entry). Data with the same superscript letter (a, b, c, or d) within a column indicate the treatments are not significantly different at *p* < 0.05.

**Table 4 materials-14-04128-t004:** Mechanical properties of paper impregnated with cottonseed protein and nanocellulose dispersions from wet strength tests.

Sample	Tensile Modulus (MPa)	Yield Strength (MPa)	Yield Strain (%)
Paper	20.77 ± 3.81 ^a^	0.34 ± 0.10 ^a^	2.51 ± 0.87 ^a^
Paper & 0.2% CNF	18.58 ± 3.36 ^a^	0.28 ± 0.12 ^a^	2.21 ± 0.89 ^a^
Paper & 1.0% CNC	21.02 ± 4.46 ^a^	0.29 ± 0.10 ^a^	2.11 ± 0.99 ^a^
Paper & 10% CSP	9.80 ± 1.95 ^b^	0.20 ± 0.04 ^b^	3.91 ± 1.00 ^b^
+0.2% CNF	10.98 ± 1.35 ^b^	0.20 ± 0.04 ^b^	4.01 ± 1.06 ^b^
+1.0% CNC	9.95 ± 1.42 ^b^	0.19 ± 0.05 ^b^	3.46 ± 0.91 ^b^

* Note: Each data point represents seven strips tested in triplicate (21 total tests per entry). Data with the same superscript letter (a or b) within a column indicate the treatments are not significantly different at *p* < 0.05.

## Data Availability

The data presented in this study are available on request from the corresponding author. The data are not publicly available due to possible proprietary concerns related to patent protection on this work.

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
