# Peer review of "Effect of Nanocellulose on the Properties of Cottonseed Protein Isolate as a Paper Strength Agent"

_materials, 2021, doi:10.3390/ma14154128_

Round 1
Reviewer 1 Report
The authors have presented wonderfully their investigation on "Effect of Nanocellulose on the Properties of Cottonseed Protein Isolate as a Paper Strength Additive". The experiment-derived products and the corresponding results are properly elaborated and explained well. The sustainability viewpoint can be interesting for the readers. Therefore, I would suggest accepting the manuscript in its current form. Only the abbreviation CNM (I think, Cellulose NanoMaterials) should be introduced in the main text in the first place of its appearance.
Author Response
Thank you for your comments. We have provide responses in the attached file.

Reviewer 2 Report
The manuscript entitled “Effect of Nanocellulose on the Properties of Cottonseed Protein Isolate as a Paper Strength Additive” studies the potential of CSP and nanocellulose as paper additives on the performance of pure cellulose paper. It is important to emphasize that the authors used pure cellulose paper in the study which differ from commercially available papers since there is no additives in it, or it is not produced in similar way like converting which in the end affects is properties. Thus, can the made dispersion react with industrial paper in different way?
As I have understood, the treatment was performed on the surface of the paper as coating? Is that correct? In this case, can it be classified as additive or coating? Additives in paper are added in the pulp during paper production.
In the whole manuscript the brackets of the references should be stated before the comma, like: “Carboxymethylated CNFs improved the wet and dry performance of prepared paper sheets when used as an additive with poly(aminoamide)-epichlorohydrin [8]. not as “Carboxymethylated CNFs improved the wet and dry performance of prepared paper sheets when used as an additive with poly(aminoamide)- epichlorohydrin.[8]”
Additional problems:
Line 22: paper products? How many paper product have you prepared and what are these? Paper is just one product. Have you made something else than paper?
Lines 33-34: “…CNCs and CNFs do not persist for extended periods of time in the
environment and are completely biodegradable.” – missing the references here
Lines 34 – 36: “Carboxymethylated CNFs improved the wet and
dry performance of prepared paper sheets when used as an additive with poly(aminoamide)-
epichlorohydrin.[8]” – of prepared laboratory paper sheets or industry? It is important to emphasize since they differ in properties and structure sometimes.
Line 93: wrong font
Line 98: if you have used soft brush how did you control the thickness?
Line 102: Have the papers before and after weight conditioned in the same conditions – temperature, humidity, pressure? Because, the paper is very hygroscopic material which weight can be affected with those parameters.
Line 149: treated with just water – please reorganize this part of sentence, it is not grammatically correct
Line 162-163: Table 1. Fist, in the page 2 you have stated that paper has thickness of 180 microns, in the table there is a slight difference presented. Please correct this. Second, the variations in thickness measurements are very large, why? Maybe because of the application with soft brush? Third, you haven’t mention how did you measure the thickness? Apparatus?
Another point is that, when presenting a Figure or Table in the manuscript, they have to be mentioned in the text. In this case, the Table 1 and Figure 1 are not connected to the text.
Lines 162 – 167: Are the results presented in Table 1 and Figure 1 the same? If are, there is no need for presentation of the same results in two ways. Please choose only one.
Line 184-185: filter paper has been mentioned here for the first time. Whatman, the paper you have used, is commonly used as filter paper. But in your case, is a pure cellulose paper. You haven’t used it for filtration. Pleas, do not use slang in the manuscript, adjust it to scientific language.
Line 204: Again, you have here Table 1. It sould be Table 2. Problem is repeating at lines 263, 279
Line 209: what does it mean “as-received paper samples”? how many of them have you used?
Lines 208 – 2016: the paragraph related to FTIR spectroscopy is minor and does not tell nothing about the samples. There should be more information related to some relevant peaks, for example those related to crystallinity of cellulose. The peaks related to cellulose as well… the images (lines in images) are blurry, and try to show a smaller area because there is too much noise. So some significant changes cannot be seen. This section should be corrected. The axis of the wavenumber should be present form 4000 – 400 as stated in experimental par (there is no 400 cm-1 in the axis)
Lines 222, 224 and 230: SEM micrographs not images
Lines 263 and 268: there is no need for presentation the same results in two different ways. Please choose one.
Author Response

(The authors gave the same response as above.)

Reviewer 3 Report
The manuscript Effect of nanocellulose on the properties of cottonseed protein isolate as a paper strength additive, describes the preparation and characterization of cottonseed protein isolate with/without cellulose nanocrystals (CNCs) and cellulose nanofibers (CNFs), used as co-additives in order to increase the dry strength of paper products.
- In my opinion, the title of this paper is not the best choice, due to the fact that the authors highlight more the changes brought by the presence of cottonseed protein adhesive, than that of nanocelluloses. In addition, even the authors point out that there is no difference between the mechanical properties of protein adhesive-treated paper and those treated with protein adhesive-nanocellulose (NCs) (pp. 6, L. 289-290), which confirms that cottonseed protein is what causes changes in the properties of paper and not NCs! The presence of NCs determines only minor changes, which may even fall within the limits of the experimental errors.
- Why do the authors use the term “additive” and not the “adhesive” for cottonseed protein? The product is a protein adhesive already studied by the authors and their data are presented in several works in the literature (references 11, 12, 15, 16, 17, 18, 19, 24)! In this sense, I ask again, what does this work bring new?
- The authors do not bring enough information in the Introduction section. There is no literature data of the achievements on the same topic with the current manuscript. Also, should the authors mention what is new about this work compared to other works in the literature, even those published by the authors? They do not argue the need to use NCs-like materials as an adhesive/additive to increase the strength of the paper.
- Reference [8] is not the proper one, due to the fact that in that study NFC is uses as an additive in paper stock composition and not as a coated additive! It must be changed with an appropriate reference!
- The method of applying the adhesive is a bit artisanal (soft brush) and does not lead to reproducible results. How can the authors demonstrate the reproducibility of the data, if the method chosen for the layer deposition does not ensure a uniform application of the surface treatment?
- Why did the authors choose to compare the treated papers with solutions of different concentrations of nanocelluloses, such as Paper & 0.2% CNF with Paper & 1.0% CNC? How do the authors highlight the influence of each type of nanocellulose (CNF and CNC) if different addition dosages and concentrations are used? What was the reasoning for choosing such different concentrations? Could be the viscosity?
- Why did the authors choose to present the variations in thickness of the samples, if these do not vary significantly? Moreover, variations in thickness can also be determined by the fact that an additive deposition process is used which cannot be rigorously controlled.
- It is not appropriate to use the same data in the table (Table 1) and in the form of a graph (Figure 1)!!! The same is observed for Table 2 and Figure 6!! The authors have to decide on a single form of data presentation!
- The paper does not contain an interpretation of the effect of the protein preparation on the resistance in dry and wet state, respectively. The differences may explain the type of bonds established between the protein and the cellulose fibers in the paper structure.
- If the authors apply the ASTM D 829-97 standard, why did they choose to use such a low rate of separation of the two grips, of only 1 mm/min, when the rate of 25 mm/min are used to test paper and cardboard samples with a constant elongation rate? Too much creep certainly alters the results!
- L. 103: “the paper width”? Could it be paper “caliper” or “thickness”? Please revise the terminology!
- L. 237: “alkali water”?
- What are FTIR investigations used for? Just to prove that the protein adhesive was deposited on the surface of the samples? Why the authors didn't prove the linkage established between protein and cellulose highlighted?
- The high increases in tensile strength that are recorded in the paper (86 and 97%, respectively) are mainly due to the low strength of the base paper on which the impregnation was made and to which it reported (a filter paper grade, designed for filtration and not characterized by high strength properties). This fact must be specifying every time when it is mentioned the increase of the tensile strength, in order not to induce a wrong image in the reader’s mind.
- p. 6, L. 193-195: The authors assumed that there are differences between paper alone and paper with CNCs, but no differences between paper treated with CNCs or treated with CNFs. How can explain the authors this assumption if the variations are very small (from 9.39 to 10.73), it is almost no differences between the samples.
None of the investigations make essential contributions to detecting the mechanism by which the strength of the paper increases and also, the contribution of each component to this increase.
The paper does not make significant contributions in the field compared to other manuscripts, even those published by the authors, but could be published in Materials journal, after substantial improvements. It would be desirable that the revised manuscript to contain the answers to all the questions formulated in the present review.
Author Response

(The authors gave the same response as above.)

Round 2
Reviewer 2 Report
Authors replied to all my questions.
Author Response
Dear Reviewer #2,
Thank you for your valuable time and comments.
Sincerely,
The Authors
Reviewer 3 Report
The manuscript materials-1289499 has been improved over the previous version.
However, I still support the idea that the formulation used by the authors is not an additive, due to the fact that it represents approximately 1/5 - 1/4 of the sample mass, which is a quantity specific to a raw material component.
The data are: 22.4 ± 6.2% for Paper & 10% CSP and 26.1 ± 4.6% for Paper & 10% CSP + 0.2% CNF, as the authors themselves mentioned (L. 215: "weight pick-up with values of approximately 26%") .
The additive used is far from the definition of additive, which usually involves adding a small amount of substances.
- Please make the correction at L. 323 and add the unit characteristic of the wavenumber: “indicates a shift from 1633 for CSP by itself to 1641 for”!!!
- Please be consistent in affirmations (L. 321-323)! It is “1641 cm-1 for the protein in contact” or “prominent absorption bands occur at 1530 and 1640 cm–1”?
- 324-325: “This 8 cm-1 shift likely suggests hydrogen bonding interactions between the amide bonds and the hydroxyl groups on the cotton fibers”?? This is an unclear and unsafe assumption! The hydrogen bonds cannot be established between chemical groups and bonds! Moreover, a shift in wavenumber does not mean that hydroxyl bonds are established!
- Please make the adequate correction to the FTIR characterization part and bring logical explanations!
Author Response
Dear Reviewer #3,
Thank you for reveiwing the revised version of our manuscript, “Effect of Nanocellulose on the Properties of Cottonseed Protein Isolate as a Paper Strength Agent” for publication in Materials. In the attached file you will find the referee reports along with our responses in blue.
Regards,
The Authors
